# Synthesis and Properties of Modified Biodegradable Polymers Based on Caprolactone

**DOI:** 10.3390/polym15244731

**Published:** 2023-12-17

**Authors:** Maria E. Fortună, Elena Ungureanu, Răzvan Rotaru, Alexandra Bargan, Ovidiu C. Ungureanu, Carmen O. Brezuleanu, Valeria Harabagiu

**Affiliations:** 1Institute of Macromolecular Chemistry “Petru Poni”, 41A Grigore Ghica Voda Alley, 700487 Iasi, Romania; fortuna.maria@icmpp.ro (M.E.F.); rotaru.razvan@icmpp.ro (R.R.); anistor@icmpp.ro (A.B.); hvaleria@icmpp.ro (V.H.); 2“Ion Ionescu de la Brad” Iasi University of Life Sciences, 3 Mihail Sadoveanu Alley, 700490 Iasi, Romania; olgutabrez@uaiasi.ro; 3“Vasile Goldis” Western University of Arad, 94 the Boulevard of the Revolution, 310025 Arad, Romania; ungureanu.ovidiu@uvvg.ro

**Keywords:** aminopropyl-terminated polydimethylsiloxane, chlorophyll, PDMS-CL copolymers, *Lypercosium esculentum*, poly-ɛ-caprolactone

## Abstract

In this paper, the synthesis and characterization of two polycaprolactone-polydimethylsiloxane (PDMS-CL) copolymers with biodegradable properties are reported. A comparative study was carried out using an aminopropyl-terminated polydimethylsiloxane macro-initiator (APDMS) with two different molecular weights. The copolymers (PDMS-CL-1 and PDMS-CL-2) were obtained by ring-opening polymerization of ɛ-caprolactone using APDMS as initiators and stannous 2-ethylhexanoate as a catalyst. The copolymer’s structures were confirmed by Fourier transform infrared spectroscopy (FT-IR), nuclear magnetic resonance (^1^H-NMR) spectra, and energy dispersion spectroscopy (EDX). Surface morphology was investigated using scanning electron microscopy (SEM) and atomic force microscopy (AFM). The hydrophobic properties of the copolymers were demonstrated by the water contact angle and water vapor sorption capacity. Additionally, biological tests were conducted on San Marzano type tomato plants (*Lypercosium esculentum*) to assess the synthesized copolymers’ susceptibility to the environment in terms of biological stability and metabolic activity. The biodegradation of PDMS-CL-1 and PDMS-CL-2 copolymers does not have a dangerous effect on the metabolic activity of plants, which makes it a convenient product in interaction with the environment.

## 1. Introduction

In the last few years, interest in the preparation of biodegradable polymers has grown due to their various properties, which offer many biomedical applications [1].

Polymers such as polyesters, polycaprolactone, polylactic acid, and poly(trimethylene carbonate) are among the most widely used polymeric biomaterials because of their biocompatibility and biodegradation [2,3]. Poly-ɛ-caprolactone (CL) is an interesting hydrophobic polymer with a semi-crystalline structure consisting of caprolactone subunits, which were formed through ring-opening polymerization, an important technique for the preparation of copolymers [4,5,6]. Moreover, poly-ɛ-caprolactone is a biodegradable synthetic polymer used in different applications in the medical field (drug delivery, wound dressing, tissue engineering), toughening agents for epoxy resins, environmentally sustainable packaging, and many others [7,8]. CL can be degraded by many microorganisms, such as several bacteria and fungi, thus making it able to undergo biodegradation [9]. Polysiloxanes represent a unique hybrid of organic and inorganic components with a specific structure that consists of a backbone formed of alternating silicon and oxygen atoms [10,11]. Polydimethylsiloxane (PDMS) is the most well-known polysiloxane. The literature offers information on many performance properties [5,6], including biocompatibility, low viscosity, and excellent characteristics such as hydrophobicity, flexibility, chemical inertness, stability at both high and low temperatures, and superior surface wettability. Considering the advantages of poly(dimethylsiloxane), it is used in a variety of medical applications and as a water treatment absorbent. Despite their interesting characteristics, polysiloxanes have limitations due to their increased cost and poor mechanical properties [12]. Thus, the use of PDMS in the preparation of copolymers demonstrates its many benefits, and it has been widely studied due to the many unique properties that emerge from the combinations of subunits [13,14]. Changes in the groups at the ends of the polydimethylsiloxane chain affect its properties. Polysiloxanes functionalized with aminoalkyl groups are known in textiles and cosmetics [15]. More studies reported on the synthesis of polydimethylsiloxane copolymers using reactive hydroxyl or amine functional oligomers that are mono- or difunctionally terminated as initiators [16,17].

It was shown in prior work [18] that CL allows the moderate development of two micromycetes, *Fusidium viride* and *Penicillium brevicompactum*, capable of biosynthesizing enzymes that accelerate biotic reactions and trigger the biodegradation of the copolymer. This destruction can be a potentially toxic action for the environment, and it is absolutely necessary to monitor the impact on the plant’s metabolism, given the conversion of the copolymer into simple substances that can modify the metabolic activity and implicitly the structure of the plant [19]. Following the decomposition of the copolymer, the nitrogen dynamics undergo evolutionary changes, confirming the process’s development with the release of organic nitrogen. In addition, the biological tests carried out (germination index, average seedling height, green and dry biomass) on *Lypercosium esculentum*, San Marzano variety tomato plants that were in direct interaction with the copolymer showed their normal growth and development, which suggests a negligible toxic effect and implicitly a compatibility of the copolymer with the environment [18]. Given the essential role of this pigment in photosynthesis and, indirectly, in the metabolism and physiological state of the plants, it is very important to quantify the total chlorophyll content of the plant leaves that came into contact with the copolymers [19]. By mixing CL with other polymers (such as polydimethylsiloxane), it is possible to synthesize copolymers or composites with modified physical, chemical, and mechanical properties [20].

The literature studies of the synthesis, characterization, and properties of PDMS-CL copolymers are limited. However, the PDMS-CL copolymer-based microspheres have unique properties and can be used to obtain biomedical devices such as scaffolds in tissue engineering and matrix resins for time-release drugs [9,19]. Polydimethylsiloxane (PDMS) has a very low glass transition temperature, which makes it a soft segment to adjust the mechanical properties of CL-based shape memory polymers (SMPs) [21]. PDMS, for instance, can be used as soft segments of varied length to tailor the mechanical properties of PCL-based polymers (acting as hard segments). By varying the segment lengths of PCL and PDMS, the mechanical properties of the obtained copolymers can be altered to achieve the desired stiffness [6].

The objective of the present study was to obtain PDMS-CL copolymers and evaluate their structure, morphology, and property behavior obtained by ring-opening polymerization. FT-IR, ^1^HNMR, and EDX spectroscopy were used to investigate the structures of the macro-initiator and synthesized copolymers. Copolymer surface properties were examined using water vapor sorption capacity and contact angle measurements, while the morphology of copolymers was investigated using SEM and AFM. The thermal behavior of the copolymers was determined via thermogravimetric analysis (TGA) and differential scanning calorimetry (DSC).

## 2. Materials and Methods

### 2.1. Materials

Octamethylcyclotetrasiloxane (D4), tetramethylamonium hydroxide pentahydrate (Aldrich, St. Louis, MO, USA), and 1,3-Bis (3 aminopropyl)-1,1,3,3-tetramethyldisiloxane were used for APDMS-1, which was synthesized according to the methods described by Fortuna et al., where a mass of approximately 3409 g/mol was obtained [9]. APDMS-2 (aminopropyl-terminated polydimethylsiloxane with a Mn of ca. 1000) was purchased from Aldrich; ɛ-caprolactone was purchased from Aldrich; and stannous octoate was obtained from Air Products. Toluene, methanol, xylene, and tetramethylammonium hydroxide were purchased from Aldrich.

Biological material: tomato plants (*Lypercosium esculentum*), San Marzano variety, vegetation period of 35 days [18].

### 2.2. Methods

FTIR spectra were recorded on a Bruker Vertex 70 device (Bruker Optics, Ettlingen, Germany). Registrations were performed at room temperature in the 400–4100 cm^−1^ range.

^1^H-NMR spectrum was investigated with a Bruker NMR spectrometer (Model DRX400, Billerica, MA, USA).

The water contact angles of the copolymers were determined with an eScope Conrad USB digital microscope and ImageJ software 1.48. A quantity of 0.5 g of copolymer was dissolved in 10 mL of chloroform, and part of it was spread on a glass slide. Then the solvent evaporated. Thin polymer films were then dried under vacuum at room temperature for 48 h. The same procedure was repeated for all samples [6].

The morphology and composition of samples were examined using scanning electron microscopy and energy dispersive X-ray spectroscopy (SEM/EDX) (Ametek, Berwyn, PA, USA).

The AFM images were obtained with a NTEGRA scanning probe microscope (NT-MDT Spectrum Instruments, Moscow, Russia). For image acquisition, the Nova v.19891 Solver software was used.

Thermogravimetric (TG) measurements and differential scanning calorimetry (DSC) analyses were performed using STA 449F1 Jupiter equipment (Netzsch Company, Selb, Germany) and a DSC 200 F3 Maia device (Netzsch, Germany), respectively. The measurements were performed in the −150 to 200 °C temperature range at a heating rate of 10 °C/min in an inert atmosphere.

The water vapor sorption capacity of the PDMS-PCL copolymers was appreciated on the basis of the sorption isotherms recorded in the dynamic regime using a fully automated gravimetric device, IGAsorp, from Hidden Analytical (Warrington, UK). Registrations were completed at 25 °C in a stream of nitrogen (250 mL/min) in the humidity range 0–90% in steps of 10% with a predetermined equilibrium time between 40 and 60 min at each step.

Chlorophyll content in leaves provides pertinent information about the physiological and metabolic states of plants. Determination of total chlorophyll content: in some plant pots (having a height of 9 cm and a diameter of 6 cm), 80 g of soil-black peat was introduced and 0.3 g of samples (PCL, APDMS-1, APDMS-2, PDMS-PCL-1, and PDMS-PCL-2) together with 3 tomato seeds [21]. The experiment was synthesized according to the method described by Fortuna et al. [18].

The total chlorophyll content was read using a CCM-200 plus chlorophyll content meter from Opti-Sciences Inc., Hudson, NH, USA. The measurements were made on the tomato plant leaves after 35 days of planting, and three samples were used for replication.

### 2.3. Macro-Initiator Synthesis-Aminopropyl-Terminated Polydimethylsiloxane (APDMS-1)

The goal of the synthesis was to synthesize an average molecular weight aminopropyl-terminated polydime-thylsiloxane (APDMS-1), which could be used as macro-initiators in the reaction with poly(ɛ-caprolactone) in a subsequent step. Appendix A illustrates the reaction for the synthesis of the polymer, which has two active amino groups at the end of the chain.

An earlier experiment [9] obtained the macro-initiator (APDMS-1) by synthesizing 1,3-bis (3 ami-nopropyl)-1,1,3,3-tetramethyldisiloxane with octamethylcyclotetrasiloxane (D4) in the presence of a tetramethylammonium hydroxide catalyst. In the first stage of the synthesis, a solution (10%) of (CH_3_)_4_NOH (2.7 mmol) in methanol was prepared in a round-bottom flask, heated by means of an oil bath under magnetic stirring, equipped with a thermometer and with connections to argon and the vacuum pump; the D4 compound (2.7 mmol) and 15 mL of toluene were added to the prepared catalyst solution; the reaction mixture was heated to 80 ± 2 °C and kept under the protection of nitrogen for 24 h. The system was then vacuum pumped to remove the solvent. Then, D4 (40 mmol), 20 mmol of 1,3-Bis (3 aminopropyl)-1,1,3,3-tetramethyldisiloxane, and 15 mL of toluene were added and kept at 120 °C under nitrogen protection for 8 h. After the equilibration reaction was completed, unreacted cyclic oligosiloxane was separated from the linear polymer using low pressure distillation. Finally, aminopropyl-terminated polydimethylsiloxane (APDMS) was synthesized. The compounds’ structures were confirmed by FT-IR and ^1^HNMR studies.

### 2.4. Preparation of the Aminopropyl-Terminated Polydimethylsiloxane-Poly(ɛ-caprolactone) Copolymers: PDMS-PCL-1 and PDMS-PCL-2

The PDMS-PCL-1 and PDMS-PCL-2 copolymers were obtained by synthesis between the aminopropyl-terminated polydimethylsiloxane as the initiator and ɛ-caprolactone by ring-opening polymerization in the presence of Sn(Oct)_2_ as the catalyst, as shown in Appendix A. The final products have a yield of 92%.

## 3. Results and Discussion

### 3.1. Structural Characterization

#### 3.1.1. Fourier Transform Infrared (FTIR) Analysis

FT-IR spectroscopy was utilized to examine the chemical composition of the copolymers and the functional groups that are responsible for the reaction between the poly-ε-caprolactone and aminopropyl-siloxane. The FT-IR spectra of the APDMS macro-initiators and the PDMS-PCL copolymers are shown in Figure 1. As observed from Figure 1, the infrared spectrum of PCL shows characteristic bands at 2936 cm^−1^ (ν_as_(CH_2_): asymmetric CH_2_ stretching), 2862 cm^−1^ (ν_as_(CH_2_): symmetric CH_2_ stretching), 1732 cm^−1^ (ν(C=O): carbonyl stretching), 1288 cm^−1^ (ν_cr_: C–O and C–C stretching in the crystalline phase), 1240 cm^−1^ (ν_as_(COC): asymmetric COC stretching), 1190 cm^−1^ (ν(OC–O): OC–O stretching), 1169 cm^−1^ (ν_as_(COC): symmetric COC stretching), and 1157 cm^−1^ (ν_am_: C–O and C–C stretching in the amorphous plane) [8,22].

Aminopropyl-terminated polydimethylsiloxane (APDMS-1 and APDMS-2) shows bands at 3393 and 3115 cm^−1^ (–NH_2_ vibration), 2963, 2964 and 2907 cm^−1^ (–CH_3_ and –CH_2_ vibrations), 1413 and 1410 cm^−1^ (δ_s_ (C–H), CH_3_), 1261 and 1259 cm^−1^ (structural vibration of Si–CH_3_), 1090, 1043, 1022 and 1020 cm^−1^ (Si–O–Si asymmetric stretching vibration), 866 and 851 cm^−1^ (δ_as_ (C–H) rocking, Si(CH_3_)_2_), 702 and 698 cm^−1^ (ν_s_ (Si–C), Si(CH_3_)_2_) [8,23]. APDMS-2 with low molecular weights did not show NH_2_ vibration, which is seen in APDMS-1 at 3115 and 3393 cm^−1^ but this is in agreement with the studies of Q. Ran [23].

PDMS-PCL-1 and PDMS-PCL-2 copolymers (Figure 2) show the presence of bands characteristic of aminopropyl-terminated polydimethylsiloxane and poly-ε-caprolactone. Thus, the asymmetric stretching vibration of the Si–O–Si bond can be seen in the wave numbers of 1097, 1036, and 1020 cm^−1^ for PDMS-PCL-1 and 1088, 1055, and 1022 cm^−1^ for PDMS-PCL-2. A high-intensity spectral band from 1732 cm^−1^ specific to caprolactone (carbonyl stretching vibration) is found in the spectrum of composites at 1724 cm^−1^ for PDMS-PCL-1 and at 1728 cm^−1^ for PDMS-PCL-2. The displacements of some wave numbers at the level of composites do not distort the fundamental structure of polysiloxane and poly-ε-caprolactone but suggest chemical interactions between the two.

Moreover, the structure of PDMS-PCL-1 and PDMS-PCL-2 copolymers was confirmed by ^1^H NMR -registered spectra presented in Appendix A. The ^1^H NMR spectra indicate that the obtained products combine the structural features of PCL and APDMS, which means that the PDMS-PCL-1 and PDMS-PCL-2 copolymers were successfully obtained.

#### 3.1.2. EDX Elemental Analysis

EDX elemental analysis was used to identify and characterize the elemental composition of PDMS-PCL-1 and PDMS-PCL-2 copolymers. From Figure 3, it can be concluded that the presence of siloxane is indicated by the peak at approximately 1.75 keV, which corresponds to the silicon atom [24].

### 3.2. Surface Morphology

#### 3.2.1. Scanning Electron Microscopy Images of PDMS-PCL-1 and PDMS-PCL-2 Copolymers

Figure 4 shows SEM images of the PDMS-PCL-1 and PDMS-PCL-2 copolymers, and different morphologies can be observed. The PDMS-PCL-1 composite is presented in the form of flat agglomerations of particles with dimensions of 20–50 microns, tightly joined together. The PDMS-PCL-2 composite is presented in the form of approximately spherical agglomerations of particles with radii of 40–50 microns. The difference in shape of the agglomerations of the two composites is probably due to the different molecular masses of the PDMS. In PDMS-PCL-2, where the PDMS has a lower molecular mass, the small lengths of the polymer chains determine windings in the form of approximately spherical balls.

#### 3.2.2. Atomic Force Microscopy (AFM) Investigations

Morphological studies were carried out via AFM investigations. AFM images (Figure 5) confirm the scanning electron microscopy (SEM). In the case of the PDMS-PCL-1 composite, the agglomerations of tightly joined particles can be clearly observed, and the surface is rough with few agglomerations on the height. In the case of the PDMS-PCL-2 composite (with a lower molecular mass), the much larger agglomerations of particles cause the appearance of large bumps in the form of hills and valleys, a phenomenon that may influence the hydrophobicity of the composite.

#### 3.2.3. Thermal Properties of the PDMS-PCL Copolymers

Figure 6 and Figure 7 show the registered TG and DTG curves of APDMS-1 (a) and APDMS-2 (b) macro-initiators and PDMS-PCL-1 (a) and PDMS-PCL-2 (b) copolymers under a nitrogen atmosphere. For the APDMS-1 (a) sample, weight loss can occur at 157 °C (with a weight loss of 21.54%), at 411 °C (with a weight loss of 27.48%), at 592 °C (with a weight loss of 41.67%), and at 699 °C, with a remaining residual mass of 5.57%. For the APDMS-2 (b) copolymer, it could also be observed that there are more weight loss steps situated at different temperatures: at 83 °C (with a weight loss of 3.33%), at 200 (with a weight loss of 8.54%), at 421 °C (with a weight loss of 34%), at 639 °C (with a weight loss of 47.72%), and at 699 °C, with a remaining residual mass of 5.81%.

According to TG analysis, the macro-initiators of APDMS-1 and APDMS-2 showed thermal stability up to approximately 135 °C and 164 °C, respectively. The next steps are associated with their decomposition. The literature [25] indicates that the first process of decomposition at low temperatures is associated with water loss. The decomposition of the copolymers and the loss of structural water are the next heat processes. There were no signs of degradation at temperatures over 400–700 °C.

One weight loss step is seen for the PDMS-PCL-1 copolymer at about 367 °C with a weight loss of 93.21%, leaving at 699.45 °C a residual mass of 6.14%; for PDMS-PCL-2 copolymer, one weight loss step is observed at about 386 °C with a weight loss of 6.85%, leaving at 699.45 °C a residual mass of 5.71%. The weight loss in the residual mass range is indicative of the material’s decomposition, with the mass residue remaining after it has been removed [9].

#### 3.2.4. Differential Scanning Calorimetry (DSC) Analysis

The DSC of the copolymers is illustrated in Figure 8 and Figure 9. The DSC measures were undertaken after the second heating scan, which was used for the elimination of residual solvents and to remove the thermal history of the polymer. The samples were heated at a rate of 10 °C per minute on a differential scanning calorimeter, ranging from −150 °C to 100 °C. DSC analyses of the copolymers confirmed the characteristic peaks associated with the melting temperatures of APDMS (low temperature) and PCL (high temperature. The first measurements (first heating) of macro-initiators demonstrate a glass transition (Tg) at −110.1 °C for PDMS-1 (Figure 8) and −121.6 °C for APDMS-2 (Figure 9). In the second heating, the glass transition was obtained at −114.2 °C and −121.9 °C.

In the case of copolymers, the first measurements (first heating) showed distinct endothermic peaks at 59.1 °C and 63.6 °C for PDMS-PCL-1 (Figure 8) and 49.7 °C for PDMS-PCL-2 (Figure 9). The melting transition of the PCL is responsible for the endothermic peaks observed in the second measurement (second heating) for PDMS-PCL-1 at 53.5 °C and for PDMS-PCL-2 at 37 °C and 42 °C. When PDMS-PCL-1 and PDMS-PCL-2 were cooled, exothermic peaks were seen at 24.2 °C and 13.6 °C, respectively.

#### 3.2.5. Contact Angle Measurement

One of the most important parameters used to measure the wettability of a material is the contact angle. By measuring the contact angle, the hydrophilicity or hydrophobicity of the copolymers was examined.

In Figure 10, the contract angle values for PDMS-PCL-1 and PDMS-PCL-2 are 107.33° and 93.46°, respectively. All the results reveal, as expected, the hydrophobic nature of the copolymers. This effect is caused by PDMS’s extremely low surface energy, which causes it to migrate to the surface and cover the majority of the copolymer’s surface [17]. In the case of the PDMS-PCL-2 composite, due to the agglomerations of spherical particles, at the macroscopic level, a porous surface will result in the absorption of water between the clumps of particles, so the contact angle was found to be smaller.

Furthermore, measurements of the water contact angle were found to be reduced but above 90 when the fiber diameter was found to be larger, which indicated a hydrophobic nature. The contact angle values confirm the SEM and AFM study.

#### 3.2.6. Dynamic Vapor Sorption Analysis/Water Vapor Behavior

The influence of the water vapor/moisture from the environment on the newly obtained PDMS-PCL copolymers was investigated using the water vapor capacity determinations in dynamic conditions employing a totally automated gravimetric instrument, the IGAsorp device (made by Hiden Analytical-Warrington, Warrington, UK). The main part of this equipment is represented by an ultrasensitive microbalance, which determines the changes in the weight of the sample as the relative humidity is varied in the sample chamber at a stable and constant temperature. The measurements are directed by user-friendly software. The PDMS-PCL copolymers were dried at 25 °C in a flux of nitrogen (250 mL/min) until their weights were in equilibrium at a relative humidity, RH, under 1%, after which the samples were deposited in a special container, i.e., a Pyrex bulb. After the drying phase, the relative humidity was continuously increased from 0 to 90%, in 10% humidity steps, with every step having a pre-established equilibrium time between 40 and 60 min, and the sorption equilibrium was reached for each step. The relative humidity was reduced, and the desorption graphs were registered [25,26].

The presence of PCL helps to maintain the structure of the copolymer and acts as a barrier for water penetration. The mass gained when the samples were placed in the dynamic vapor sorption equipment was due to the moisture penetration into the copolymer matrix/backbone. The porous structure allows for the permeability of the moisture in the copolymer samples. In agreement with the results obtained for the contact angle of the samples PDMS-PCL-1 (107.33°) and PDMS-PCL-2 (93.46°), respectively, the values of the water vapor sorption capacity are PDMS-PCL-1, 0.79%, and PDMS-PCL-2, 1.89%.

The sorption/desorption isotherms are depicted in Figure 11. Looking at the shapes of these curves, which can be associated with type V isotherms according to the IUPAC classification, this type of isotherm with hysteresis can be interpreted as being representative of the porous surfaces and characteristic for hydrophobic material or very low hydrophilic material. Important information about the surface of the PDMS-PCL copolymers can be obtained from the graphs of the sorption/desorption isotherm: reduced water vapor sorption at low values of relative humidity, RH (0 to 10%), sometimes moderate sorption at intermediate values of RH, and a high increase in water sorption at RH values close to 100% [26].

The Brunauer–Emmett–Teller kinetic model (BET, Equation (1)) was used to calculate the specific surface area (Table 1) by modeling the sorption isotherms measured in dynamic conditions.
(1)$$W=\frac{W_{m}CRH}{\left(1-RH\right)\left(1-RH+CRH\right)}$$

The parameters involved are weight of sorbed water (W), weight of water forming a monolayer (Wm), sorption constant (C), and relative humidity (RH).

The average pore sizes also influence, in a special way, the sorption capacity of the PDMS-PCL copolymers. Using the Barrett, Joyner, and Halenda model (BJF, Equations (2) and (3)), and taking into consideration the calculation methods for cylindrical pores, the average pore size, rpm (Table 1), was calculated. This method uses the desorption branch of the isotherm. This desorbed moisture content is due either to the evaporation of the liquid core or to desorption of a multilayer. Pore size distribution is outlined as the distribution of pore volume. The association between pore volume and gas uptake can be defined if we know the density of the adsorbed phase. The first assumption of mesopore size analysis is that the phase is equivalent to the liquid phase of the adsorbate.
(2)$$V_{liq}=\frac{n}{100\rho_{a}}$$
(3)$$r_{pm}=\frac{2V_{liq}}{A}$$
where *V_liq_* is the water volume, *n* is the absorption percentage, *ρ_a_* is the adsorbed water density, and *A* is the specific surface area evaluated by the BET method.

The values of the surface parameters determined using the BET kinetic model demonstrated the hydrophobic-porous nature of the PDMS-PCL copolymers and can be used for further applications (biomedical, environmental).

#### 3.2.7. Biological Stability: Total Chlorophyll Content

After a vegetation period of 35 days, the tomato plants (Figure 12) were analyzed from the point of view of the total chlorophyll content, considering that this chelated compound is responsible for the photochemical reactions during the photosynthesis process and for the good growth and development of the plant.

Appendix A shows the total chlorophyll content, expressed in CCl units (% transmittance at 931 nm/% transmittance at 653 nm) for tomato plants in contact with five polymeric substances: PCL, APDMS-1, APDMS-2, PDMS-PCL-1, and PDMS-PCL-2, in relation to the control sample at the end of the vegetation period (35 days).

Analyzing experimental data allows observation of the chlorophyll pigments in the control plants and those developed in the presence of PCL close to and above those found in plants developed with APDMS-1 and APDMS-2, or PDMS-PCL-1 and PDMS-PCL-2, respectively. In addition, the presence of PCL in PDMS-1 and PDMS-2 contributes to a slight increase in the chlorophyll content compared to APDMS-1 and APDMS-2, respectively. It appears that PCL degradation has little effect on photosynthetic activity and the accumulation of assimilated compounds required for biological and metabolic processes during the vegetation period of the plants, indicating the copolymer’s susceptibility to the environment and the minimal stress induced on the plants.

## 4. Conclusions

Using ε-caprolactone and aminopropyl-terminated polydimethylsiloxane as macro-initiators with two distinct molecular weights (APDMS), PDMS-PCL-1 and PDMS-PCL-2 copolymers were synthesized. The structures were experimentally confirmed using IR, ^1^H-NMR, SEM, and EDX.

The copolymers obtained displayed two-enthalpy fusion relative to PDMS and PCL block polymers. Important facts were observed concerning the PCL crystallization evolution as a function of the PDMS content in these copolymers. The moisture sorption and desorption behavior of the copolymers was studied, and the values and the curve of the isotherms were in agreement with the results obtained for the contact angles of the samples PDMS-PCL-1 (107.33°) and PDMS-PCL-2 (93.46°) and the water vapor sorption capacities of PDMS-PCL-1 (0.79%) and PDMS-PCL-2 (1.89%), and that the isotherms have the shape and curve of the hydrophobic and porous materials. This fact can be explained by the low surface energy of PDMS, which is able to cover nearly the entire surface of the copolymers and move at the surface.

The fact that the total chlorophyll content of tomato plants grown in the presence of CL is slightly lower than, but close in value to, that obtained in the case of control plants leads to the idea that the biodegradation of the copolymers does not have a major disruptive effect on the metabolic activity of the plants, which makes it an acceptable product in interaction with the environment, thus widening its range of applications.

## Figures and Tables

**Figure 1 polymers-15-04731-f001:**
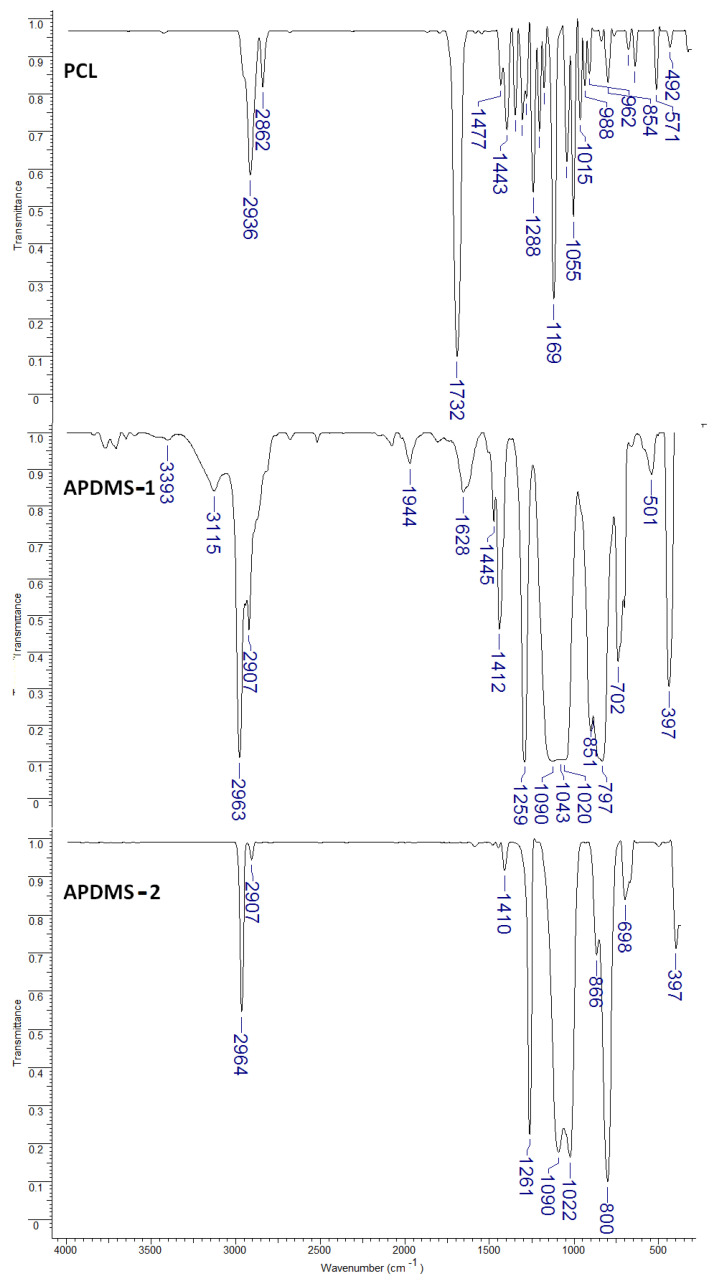
FTIR spectra for PCL, APDMS−1, and APDMS−2.

**Figure 2 polymers-15-04731-f002:**
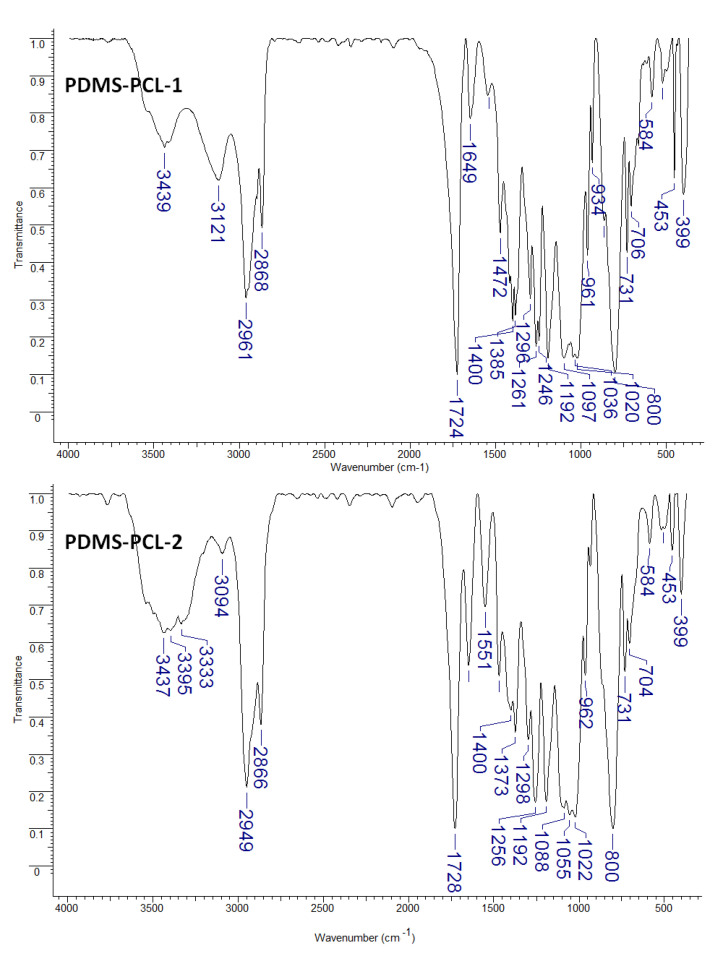
FTIR spectra for APDMS-PCL-1 and APDMS-PCL-2 composites.

**Figure 3 polymers-15-04731-f003:**
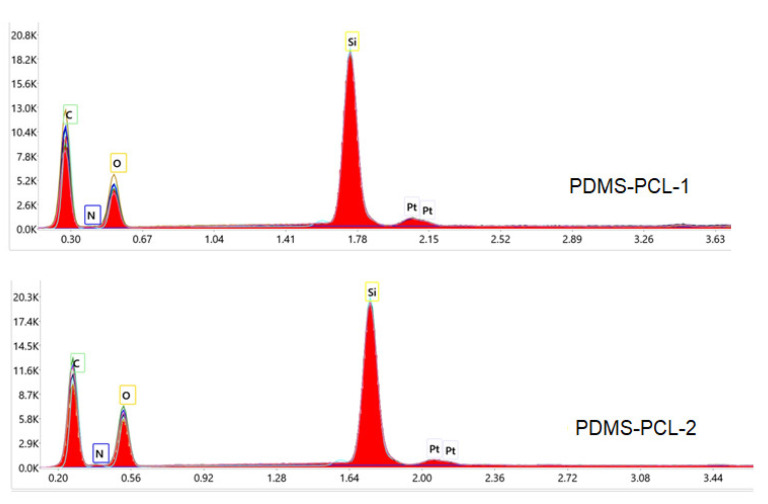
EDX spectra of PDMS-PCL-1 and PDMS-PCL-2 copolymers.

**Figure 4 polymers-15-04731-f004:**
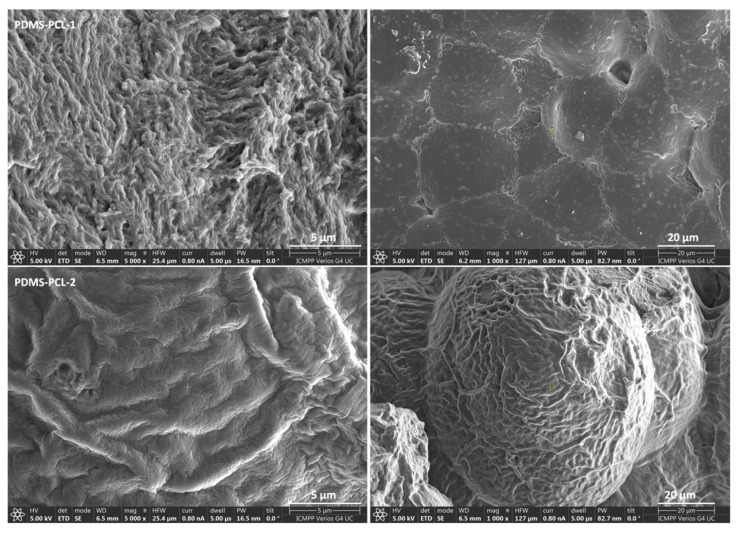
SEM images of PDMS-PCL-1 and PDMS-PCL-2 sample.

**Figure 5 polymers-15-04731-f005:**
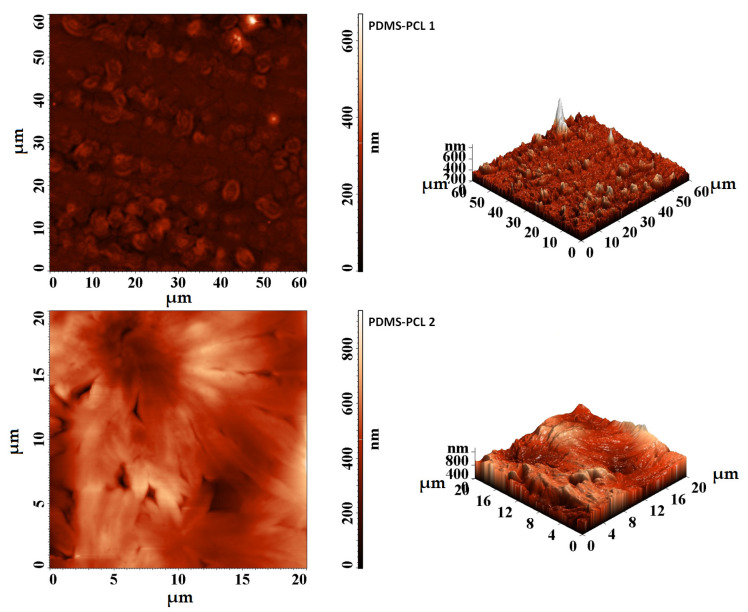
AFM images of PDMS-PCL−1 and PDMS-PCL−2 samples.

**Figure 6 polymers-15-04731-f006:**
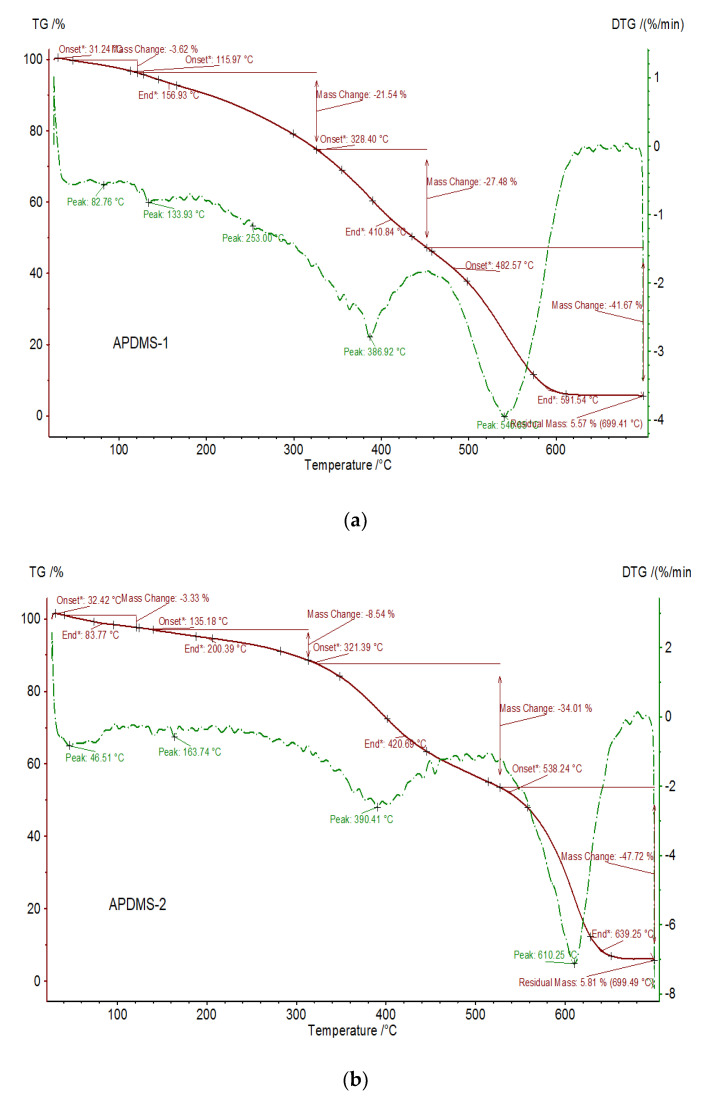
TG (red) and DTG (green) curves of APDMS−1 (**a**) and APDMS−2 (**b**) macro-initiators.

**Figure 7 polymers-15-04731-f007:**
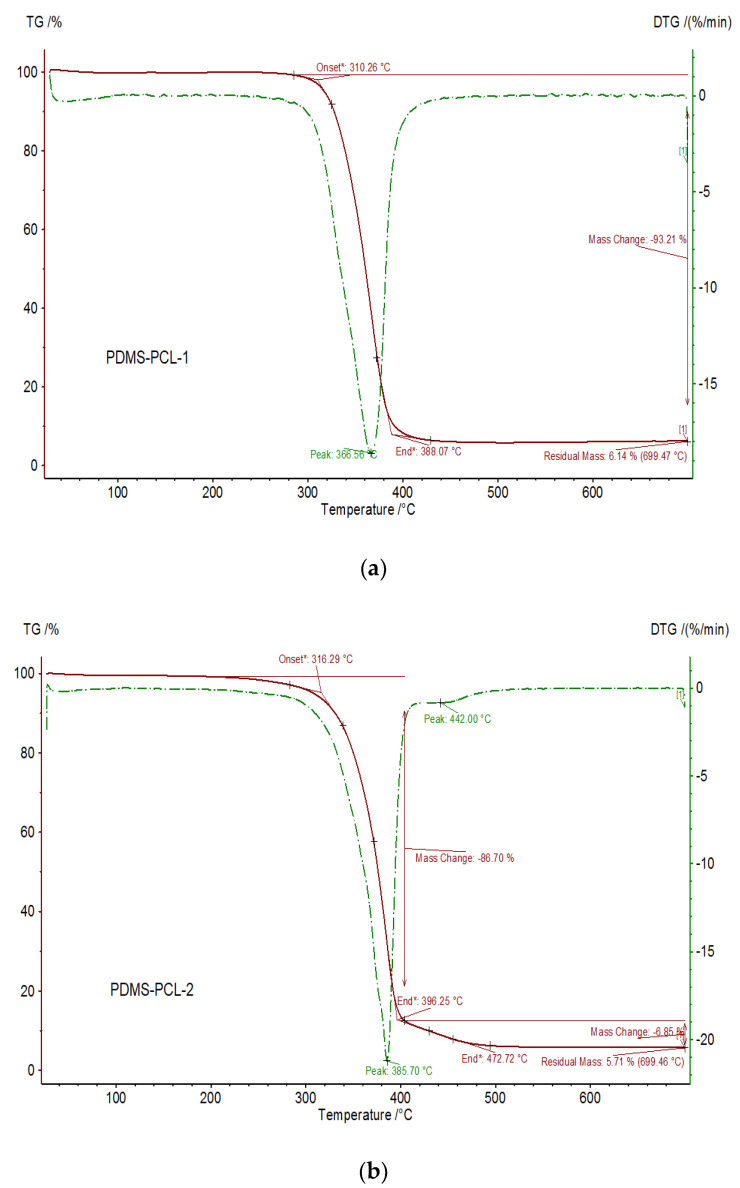
TG (red) and DTG (green) curves of PDMS−PCL-1 (**a**) and PDMS-PCL−2 (**b**) copolymers.

**Figure 8 polymers-15-04731-f008:**
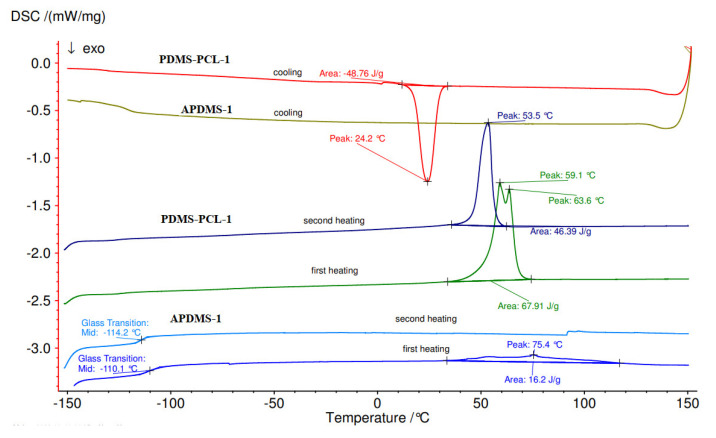
DSC curves of APDMS−1 and PDMS-PCL−1.

**Figure 9 polymers-15-04731-f009:**
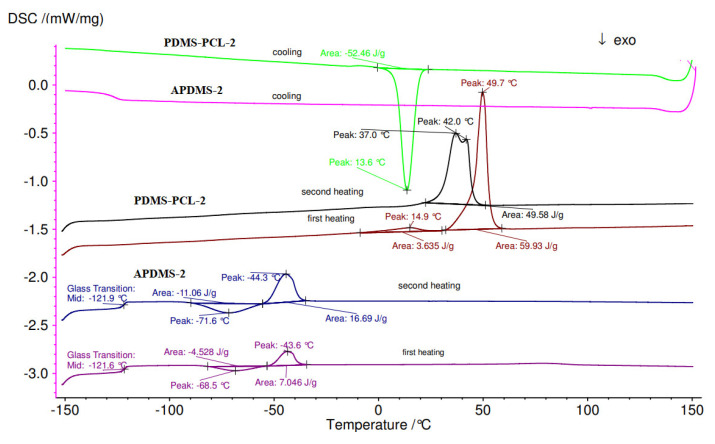
DSC curves of APDMS-2 and PDMS-PCL-2.

**Figure 10 polymers-15-04731-f010:**
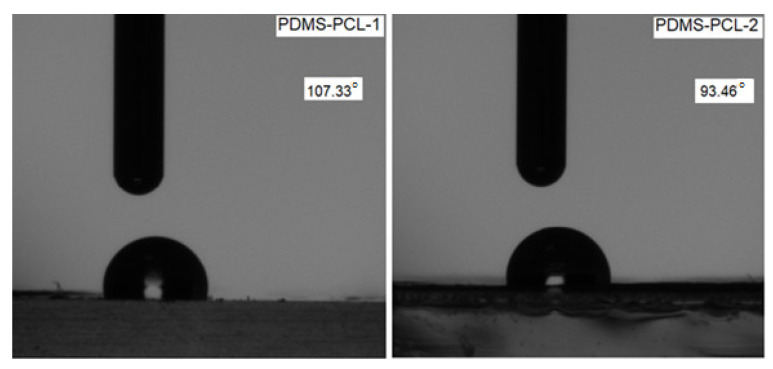
Water contact angle value.

**Figure 11 polymers-15-04731-f011:**
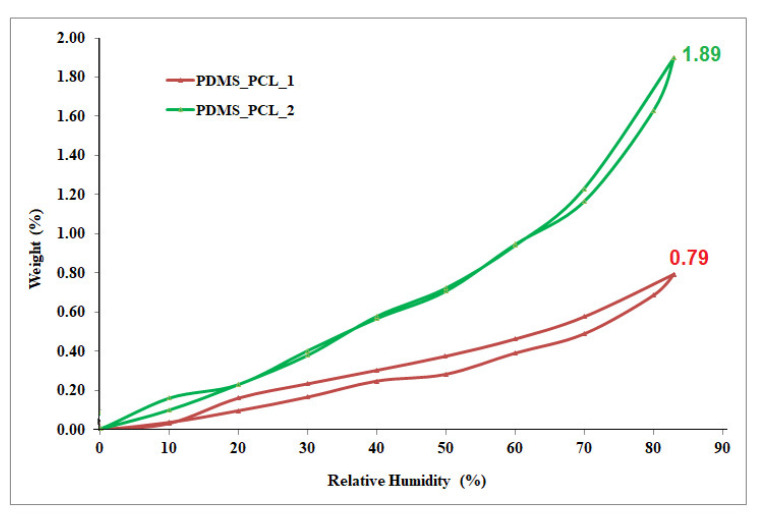
Sorption/desorption isotherms for the PDMS-PCL copolymers.

**Figure 12 polymers-15-04731-f012:**
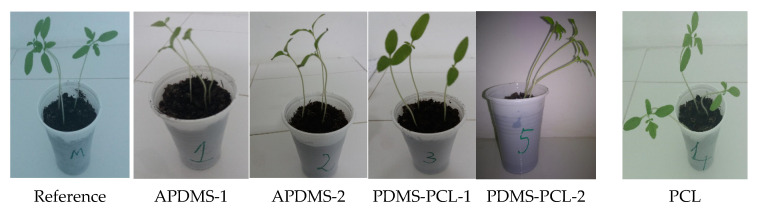
Tomato plants after 35 days of planting.

**Table 1 polymers-15-04731-t001:** The particularities of the surface determined starting from the adsorption/desorption isotherms: moisture sorption capacity; W, final weight; *r_pm_*, medium pore size and BET data for the PDMS-PCL copolymers.

Sample	W (%)	$r_{pm}$ (nm)	BET Data *	
			Area (m^2^/g)	Monolayer (g/g)
PDMS-PCL-1	0.7940	2.01	7.879	0.0022
PDMS-PCL-2	1.8993	1.26	30.007	0.0085

* Calculated based on desorption branch of the isotherm (recorded up to a relative humidity of 40%).

## Data Availability

Data are contained within the article and Appendix A.

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
