# Peer review of "Synthesis and Properties of Modified Biodegradable Polymers Based on Caprolactone"

_polymers, 2023, doi:10.3390/polym15244731_

Round 1
Reviewer 1 Report
Comments and Suggestions for Authors
The article of Fortună et al. is devoted to the synthesis and study of the properties of the biodegradable polymers based on caprolactone. The authors used a variety of physicochemical methods to analyze the properties of the copolymers, and also studied the effect of adding copolymers to soil on the chlorophyll content of tomatoes.
Authors are required to revise the paper before it can be considered for publication in Polymers.
Comments:
1) the polycaprolactone sample used in this work is a commercial product, but the molecular weight of this compound is not indicated
2) the structure of the PDMS-PCL copolymer presented in SI is incorrect - polyester has turned into polyketone
3) polycaprolactone, according to the literature, can be either a linear molecule or a macrocycle. Based on the reaction scheme presented in the SI, the authors assume that they are working with a macrocyclic polycaprolactone sample, do the authors have confirmation that this is indeed the case? It is also interesting why the authors call the process they described for obtaining a copolymer as ring-opening polymerization, while the reaction scheme shows the condensation of two polymers.
4) As sources, the authors use APDMS with a molecular weight of 4700 g/mol and 1000 g/mol, while the condensation products of APDMS with PCL have weights of 9700 and 1900 g/mol, respectively. What, according to the authors, is the reason for the different increase in molecular weight given the use of the same PCL sample?
5) in addition to the previous paragraph:
Integrating the NMR signals of the PDMS-PCL sample will determine whether there are actually 2 molecules of the original PCL for every APDMS molecule, as shown in the reaction scheme in the SI, or whether the reaction proceeds somewhat differently.
6) since PDMS-PCL is a biodegradable polymer and its degradation products can affect plants, the authors studied the chlorophyll content in tomato samples grown in the presence and absence of polymer samples in the soil. However, the article does not say anything about whether biodegradation of these polymers occurred during the incubation period of 35 days.
7) To determine the chlorophyll content in tomatoes, 3 samples were used, what is the spread of the obtained values?
Is the difference between the values of 10 and 8 CCl given in Table 2 reliable?
Reviewer 2 Report
Comments and Suggestions for Authors
While the study appears to be sound, and clearly reflects the novelty of research. The quality of paper is good and has potential of being accepted, but some important points need to be fixed before moving forward.
1. Abstract should be more explicit. Authors should include more results and data in the abstract.
2. Novelty of the work should be highlighted in the Introduction. Many paragraphs are only 2 or 3 sentences; please revise them to 4 or 7 sentences.
3. The literature review needs to be substantially enhanced by referring to published reviews on sustainability.
3. The language needs to be improved throughout the manuscript. At several places, the language used is difficult to understand. Authors should read the whole manuscript carefully.
I advise the authors work with a writing coach or copyeditor to improve the flow and readability of the text.
4. Explain the figures in a proper manner depicting the usage of the particular figure in that context so that it could be easier to relate.
5. Quality of images is poorly presented. In some figures, functional peaks should be marked.
Comments on the Quality of English LanguageAfter revision can be accepted
Round 2
Reviewer 1 Report
Comments and Suggestions for Authors
the authors partially made changes according to my comments in the first review, but Scheme 1 from the SI became even more incorrect. So the ester still miraculously turns into a ketone, in addition, the authors use polycaprolactone in their work, and in Scheme 1 they depict the monomeric unit of caprolactone.
Also, the authors did not answer anything regarding the NMR spectroscopy data. According to the integrated intensities in the NMR signal, what is the ratio of coprolactone and dimethylsiloxane units?
In addition, I recommend that the authors provide Table 2, from the “responses to reviewer” file in the SI
Reviewer 2 Report
Comments and Suggestions for Authors
Now can be accepted
Author Response
We would like to thank the reviewer for accepting.